# Effects of Acute Hypobaric Hypoxia Exposure on Cardiovascular Function in Unacclimatized Healthy Subjects: A “Rapid Ascent” Hypobaric Chamber Study

**DOI:** 10.3390/ijerph19095394

**Published:** 2022-04-28

**Authors:** Sigrid Theunissen, Costantino Balestra, Sébastien Bolognési, Guy Borgers, Dirk Vissenaeken, Georges Obeid, Peter Germonpré, Patrick M. Honoré, David De Bels

**Affiliations:** 1Environmental, Occupational, Aging (Integrative) Physiology Laboratory, Haute Ecole Bruxelles-Brabant (HE2B), 1160 Brussels, Belgium; sebastien.bolognesi@hotmail.fr; 2Physical Activity Teaching Unit, Motor Sciences Department, Université Libre de Bruxelles (ULB), 1050 Brussels, Belgium; 3DAN Europe Research Division (Roseto-Brussels), 1160 Brussels, Belgium; 4Hypobaric Centre, Queen Astrid Military Hospital, 1120 Brussels, Belgium; guy.borgers@telenet.be (G.B.); dirk.vissenaeken@gmail.com (D.V.); 5Military Hospital Queen Elizabeth, 1120 Brussels, Belgium; go@icardio.be (G.O.); pgermonpre@gmail.com (P.G.); 6Department of Intensive Care Medicine, CHU-Brugmann, 1020 Brussels, Belgium; patrick.honore@chu-brugmann.be (P.M.H.); david.debels@chu-brugmann.be (D.D.B.)

**Keywords:** hypobaric, nitric oxide, vascular reactions, breathing, extreme environments, human, permissive hypoxemia

## Abstract

**Background:** This study aimed to observe the effects of a fast acute ascent to simulated high altitudes on cardiovascular function both in the main arteries and in peripheral circulation. **Methods:** We examined 17 healthy volunteers, between 18 and 50 years old, at sea level, at 3842 m of hypobaric hypoxia and after return to sea level. Cardiac output (CO) was measured with Doppler transthoracic echocardiography. Oxygen delivery was estimated as the product of CO and peripheral oxygen saturation (SpO_2_). The brachial artery’s flow-mediated dilation (FMD) was measured with the ultrasound method. Post-occlusion reactive hyperemia (PORH) was assessed by digital plethysmography. **Results:** During altitude stay, peripheral oxygen saturation decreased (84.9 ± 4.2% of pre-ascent values; *p* < 0.001). None of the volunteers presented any hypoxia-related symptoms. Nevertheless, an increase in cardiac output (143.2 ± 36.2% of pre-ascent values, *p* < 0.001) and oxygen delivery index (120.6 ± 28.4% of pre-ascent values; *p* > 0.05) was observed. FMD decreased (97.3 ± 4.5% of pre-ascent values; *p* < 0.05) and PORH did not change throughout the whole experiment. Τhe observed changes disappeared after return to sea level, and normoxia re-ensued. **Conclusions:** Acute exposure to hypobaric hypoxia resulted in decreased oxygen saturation and increased compensatory heart rate, cardiac output and oxygen delivery. Pre-occlusion vascular diameters increase probably due to the reduction in systemic vascular resistance preventing flow-mediated dilation from increasing. Mean Arterial Pressure possibly decrease for the same reason without altering post-occlusive reactive hyperemia throughout the whole experiment, which shows that compensation mechanisms that increase oxygen delivery are effective.

## 1. Introduction

Altitude exposes the body to different kinds of environmental constraints, inducing a cascade of physiological adaptations in humans [1]. Indeed, changes in response to the lower partial pressure of oxygen include an increase in cardiac output due to increased sympathetic activity [2]. Hypoxia also induces a decrease in free radicals known to interfere with endothelial function [3] and a response from the central respiratory center: the hypoxia ventilatory response, or HVR [2]. This leads to an hypocapnic hyperventilation, which in turn stimulates the kidney to excrete more bicarbonate to compensate alkalosis.

Insufficient or inadequate adaptation can lead to symptoms such as headache, nausea, vertigo, fatigue, and cognitive disorders called acute mountain sickness [4,5]. This may partake in the appearance of high altitude-related illnesses, even after an acute exposure to hypobaric hypoxia [6].

The “Aiguille du Midi” in the Chamonix valley is one of the most visited high-altitude sites in the French Alps. Each year, more than 500,000 visitors come to admire some of Europe’s highest summits, reaching an altitude of 3842 m by taking a twenty-minute cable car ride.

The literature describes the effects of altitude after a night, a week, or longer periods of stay, but seldom the acute effects after a stay of only several hours. The objective of this research is to observe whether a short exposure to hypobaric hypoxia, comparable to the ascent to the “Aiguille du Midi”, could influence cardiovascular function for unacclimatized tourists. A study on the influence of a stay of several hours at a simulated high altitude on cognitive function has already been published by our group [7].

## 2. Materials and Methods

This was a cross-sectional prospective study in the hypobaric chamber of the Queen Astrid Military Hospital in Brussels, Belgium. The experimental protocol is represented in Figure 1. All experimental procedures were conducted following the Declaration of Helsinki [8] and were approved by the Academic Ethical Committee of Brussels (Brussels Alliance for Research and Higher Education, B200-2013-127). Written informed signed consent was received from all the volunteers enrolled. An ENT examination was also performed on all participants before the start of the study.

### 2.1. Study Population

All methods and potential risks were explained in detail to the participants. After written, informed consent, 17 subjects were enrolled in the study. They were eligible if they (1) were in good physical shape and (2) were aged between 18 and 50 years old. Exclusion criteria were (1) smoking, (2) high-level athletes, (3) known arterial hypertension, (4) any respiratory, cardiovascular, neurological or chronic diseases, and (5) treatment with any cardiovascular medication. No antioxidant nutrients, i.e., dark chocolate, red wine, or green tea, were permitted 8 h preceding and during the study. The subjects were also asked not to dive 48 h before the experiment and not to fly within 72 h before the experiment.

### 2.2. Experimental Protocol

The study took place in the Queen Astrid military hospital in Neder-Over-Heembeek (Brussels Region). The experiments were carried out over 2 days with 2 groups of 8 and 9 subjects. A climb in altitude was simulated by means of a hypobaric chamber to reach a pressure comparable to the Aiguille du Midi (3842 m). The ascent and descent took place gradually for 20 min. No physical exercise was performed during the whole procedure and participants were comfortably seated in a pilot seat which is the standard seat in the hypobaric chamber. A physician monitored individuals during the whole procedure for symptoms related to hypoxia, i.e., headache, nausea, vertigo, tiredness. No supplementary oxygen was provided unless there were clinical signs of hypoxia. Peripheral oxygen saturation (SpO_2_) and heart rate (HR) were continuously monitored (every participant had a monitor above their seat continuously displaying their personal HR and SpO_2_).

Peripheral oxygen saturation (SpO_2_), heart rate, arterial blood pressure, and cardiac and vascular parameters were measured at the following time intervals: before the rise in altitude (baseline), one hour after arrival at 3842 m and 1 h after the return to normal pressure conditions. (Figure 2).

### 2.3. Measurements

#### 2.3.1. Oxygen Saturation and Heart Rate

Blood oxygen saturation (SpO_2_) and heart rate in beats per minute (bpm) were measured in the finger through an oximeter (Hand-Held Pulse Oximeter—CMS60 F—Contec Medical Systems Co., Ltd., Qinhuangdao, China) including an infrared light probe placed on the subject’s finger. It was expressed in %. The measure took a few seconds.

#### 2.3.2. Cardiac Parameters

Using Doppler transthoracic echocardiography (transducer Mindray M7, Mindray Bio-Medical Electronics, Shenzhen, China), we measured the left ventricular outflow tract (LVOT) and blood velocity time integral (VTI) before the ascent. Measurement of VTI was repeated after a stay at 3842 m and 1 h after return to baseline level. All measurements were stored and analyzed offline. Three consecutive velocity curves were measured, and the average VTI was calculated. Using the continuity (continuity equation) principle, we calculated the stroke volume and the cardiac output according to the following usual formula: Q = (CSA × VTI) × HR, where CSA= LVOT r^2^ × π (CSA is the valve orifice cross sectional area, r is the valve radius, VTI is the velocity time integral of the trace of the Doppler flow profile and HR is the heart rate). As the result of the formula can be altered by different parameters, of which one is (de)hydration (which can significantly interfere with cardiac output), a quick subchondral view for assessment of the inferior vena cava diameter (qualitative evaluation) was performed; this led to detection of 2 participants with clinically relevant dehydration. As the calculation could not be accepted, these two were therefore not considered for these measurements.

Oxygen delivery is formally calculated using the hemoglobin oxygen saturation, dissolved oxygen content in arterial blood and cardiac output (CO). Given that dissolved oxygen content contribution to oxygen delivery value is minimal in this setting, we eliminated this variable. Furthermore, we were interested in the changes in oxygen delivery, and we considered that hemoglobin concentration remained unchanged during the study period. Therefore, to evaluate oxygen delivery changes, we used the product of cardiac output and peripheral O_2_ saturation (DO_2_ index).

##### Systemic Vascular Resistances

Systemic Vascular Resistances were calculated through the following formula: SVR = (MAP − CVP/CO) × 80 and expressed in dyne.sec.cm^−5^, where SVR is systemic vascular resistances, MAP is mean systemic arterial pressure, CVP is central venous pressure and CO is the cardiac output [9]. CVP is near zero in normal subjects and has not been considered for SVR calculations.

#### 2.3.3. Vascular Parameters

##### Blood Pressure

Systemic arterial blood pressure was measured after a 5 min rest in a supine position by an automated measuring device (Omron M5-I, Omron Healthcare, Hoofddorp, The Netherlands). Both systolic (SBP) and diastolic (DBP) blood pressure were measured and expressed in mmHg. Mean arterial blood pressure (MAP) was calculated with the following formula: MAP= (SBP + 2 × DBP)/3 [10]. All pressures were expressed in mmHg.

##### Flow Mediated Dilation (FMD)

FMD, an established measure of the endothelium-dependent vasodilation mediated by nitric oxide (NO) [11], was used to assess the effect of altitude on the main conduit arteries. Subjects were at rest for 10 min in a supine position before the measurements were taken. Brachial artery diameter was measured by means of a 5.0–10.0 MHz linear transducer using a Mindray DP-30 digital diagnostic ultrasound system immediately before and one minute after 5 min ischemia induced by inflating a cuff placed on the forearm to 180 mmHg, as described previously [12].

All ultrasound assessments were performed by an experienced operator (ST), with more than 100 scans/year, which is recommended to maintain competency with the FMD method [13].

On the images chosen for analysis, the boundaries for diameter measurement were identified manually with an electronic caliper (provided by the ultrasonography software) in a threefold repetition pattern to calculate the mean value.

FMD was calculated as the percentage increase in arterial diameter from the resting state to maximal dilation (post-occlusion diameter/pre-occlusion diameter * 100).

##### Post-Occlusive Hyperemia (PORH)

The relative dilation of small arteries was measured by post-occlusion reactive hyperemia (PORH). This technique was recently demonstrated as useful in measuring peripheral vascular function [14]. A plethysmographic probe (Cardiovarisc, FLOMEDI, Belgium) was placed on the index finger of both hands during the entire FMD procedure., The amplitude tracing of the two fingers was recorded. During cuff inflation, flow is occluded, and it is restored with values above baseline after cuff release (hyperemic period). In the contralateral control finger, flow continues throughout, and pulse amplitude undergoes minimal changes. In this test, the response of the pulse wave amplitude to hyperemia was calculated from the hyperemic fingertip as the ratio of the post-deflation pulse amplitude to the baseline amplitude as described in Kuznetsova et al. [14]. Photoplethysmography works by emitting an infrared light at a wavelength of 940 nm to illuminate the skin and measuring the amount of light reflected with a photodiode, which converts it into an electrical current. The change in light absorption reflects the path length that the light must travel in the bloodstream and therefore the dilation of the artery. The pulse trace was displayed and recorded.

### 2.4. Statistical Analysis

Statistical analyses were conducted using GraphPad Prism 7 (La Jolla, CA, USA). Data are given as a percentage of pre-ascension values. The difference between the percentage of pre-ascension values and 100% was compared by a two-tailed one-sample t test when normality of the sample was reached, as assessed by the d’Agostino and Pearson test. Otherwise, the non-parametric Wilcoxon Rank Sum test was used. Comparison between parameters at altitude and after return to sea level was achieved by means of a paired t-test or Wilcoxon matched-pairs signed rank test where appropriate. Comparisons between absolute values were performed by repeated measures one-way ANOVA or Friedman test when normality was not assumed. Significance level was set at *p* < 0.05. All data are presented as mean ± standard deviation (SD). Sample size was calculated by setting the power of the study at 95% and assuming that variables would have been affected to a similar extent as that observed in our previous studies [15].

## 3. Results

### 3.1. Study Population

Seventeen male volunteers were enrolled in the study (age: 26.1 ± 7.8 years, between 20 and 50 years; height: 179 ± 6.5 cm and weight: 77.9 ± 8.7 kg).

### 3.2. Environmental Conditions

The temperature in the hypobaric chamber was 23 °C and humidity was 32%. There was no wind and no radiation

### 3.3. Hypoxia-Induced SpO_2_ Changes

At 3842 m, peripheral oxygen saturation was significantly decreased (84.9 ± 4.2% of pre-ascent values; *p* < 0.001). None of the volunteers presented any hypoxia-related symptoms at any time of the study.

### 3.4. Cardiac Parameters

We observed an increase in heart rate between sea level and 3842 m (125.4 ± 17.04% of pre-ascent values; *p* < 0.001), returning to control values 1 h after the descent (100.5 ± 15.88% of pre-ascent values, *p* > 0.05).

Cardiac output (143.2 ± 36.2% of pre-ascent values; *p* < 0.001) and oxygen delivery index (120.6 ± 28.0; *p* < 0.05) also increased at 3842 m compared to pre-ascent values. One hour after returning to normobaric conditions, all returned to values comparable to those before the ascent (*p* > 0.05 compared to pre-ascent values) (Figure 3).

Systemic vascular resistances decreased at 3842 m (70.9 ± 17.0% of pre-ascent values; *p* < 0.001) and returned to initial values 1 h after the descent.

### 3.5. Blood Pressure

The mean arterial pressure decreased at 3842 m compared to sea level values (93.1 ± 6.5%; *p* < 0.001% of pre-ascent values), and this is the only parameter that still remained reduced one hour after the return to normobaric conditions (95.9 ± 6.5% of pre-ascent values).

### 3.6. Vascular Parameters

Pre-occlusion diameters increased at 3842 m compared to baseline values (105.5 ± 4.7% of control values; *p* < 0.001) and returned to basline one hour after return to normobaric conditions (101.2 ± 3.6% of baseline values). Conversely, flow-mediated dilation (FMD) decreased at 3842 m compared to pre-ascent values (97.27 ± 4.5% of control values; *p* < 0.05) and returned to pre-ascent values 1 h after return to normobaric conditions (100.0 ± 5.3% of control values; *p* > 0.05).

Post-occlusive reactive hyperemia (PORH) did not change throughout the experimentation (120.5 ± 51.9% and 96.2 ± 32.6% of pre-ascent values at 3842 m and 1 h after return to normobaric conditions, respectively).

Results (absolute values) are presented in Table 1.

## 4. Discussion

The main findings of this study can be summarized as follows: (1) moderate acute hypobaric hypoxemia in healthy volunteers did not cause any clinical signs of hypoxia and increased the cardiac output and oxygen delivery, (2) moderate acute hypobaric hypoxemia did not induce peripheral vasodilation, with a decrease in FMD, and (3) almost all changes return to baseline values early after return to seal level oxygen pressures [16,17].

Increasing cardiac output and peripheral vasodilation are two major compensatory mechanisms for hypoxia [18]. Both of the mechanisms aim to increase oxygen delivery to the periphery. In a clinical setting, acute hypoxemia is an emergency that can lead to tissue hypoxia and, if not compensated, cell death. Oxygen administration with higher than the ambient fraction of oxygen is the first-line treatment of these patients. In clinical practice, oxygenation targets for adult patients in the ICU are highly variable [19,20]. A multicenter study demonstrated no differences in mortality between low and high targets of oxygenation [21]. These findings lead several clinicians to accept lower levels of oxygen targets [22]. Nevertheless, given that the capacity to compensate hypoxemia is highly variable across critically ill patients (with possible impairment of nitric oxyde bioavailability), this strategy may lead to hypoxia in several cases [23,24]. Therefore, understanding physiologic responses to hypoxemia is mandatory for avoiding hypoxia, and this may help monitor critically ill patients and adapt oxygen targets [25].

These study results are on track with previous findings that in healthy individuals, acute hypoxemia to peripheral saturation between 80–90% is not usually associated with severe clinical symptoms [26]. An increase in oxygen delivery can putatively explain this adjustment. In this study, we found that the increase in cardiac output compensated greatly the decrease in oxygen saturation and even resulted in an increase in oxygen delivery. Similar to previous studies [27], we observed an increase in heart rate during acute hypoxemia that can be explained by sympathetic system activation [28]. Nevertheless, in our study, the heart rate increase only partly explains the increase in the cardiac output as we also observed an increase in the stroke volume. This apparent discrepancy with our previously published results can be explained by the fact that in our previous experiment we investigated a longer duration of hypoxemia, allowing enough time to adapt [11]. Therefore, the results of this study extend our knowledge on the cardiovascular effects of moderate hypoxemia.

At the vascular level, we observe a decrease in systemic vascular resistance. This leads to a decrease in arterial pressure and could also be the explanation for the increase in pre-occlusion diameters. Since flow-mediated dilation is the ratio of pre- to post-occlusion diameters, high pre-occlusion diameters already tend to reduce the FMD value.

Hypoxia also increases the expression and activity of Rho-kinase [29]. Rho-kinase facilitates the downregulation of the eNOS expression induced by hypoxia [29], thus reducing the endothelial production of NO. Furthermore, hypoxia increases endothelin-1 (ET-1) levels [30], impacting the eNOS inhibition via the Rho-kinase activation. This could explain the decrease in the FMD, which indicates a decrease in NO production from the endothelium. Similar observation was shown after a series of apneas by breath-holding divers [31].

Very recent data show an FMD increase during hypoxia [16], but this study used a very short hypoxia (20 min); during a longer hypoxic exposure (as in the present study), more hydroxyl radicals are endogenously produced [32]. These radicals can scavenge NO and concomitantly reduce FMD, as presented in our results. Therefore, we conclude that moderate hypoxemia without signs of hypoxia has no vasodilatory effects but may cause a decrease in endothelial nitric oxide production, opposing the already known mechanism of oxygen related vasoconstriction [33,34].

Our results are in agreement with previous studies showing that oxygen deprivation results in a significant increase in the expression of iNOS due to the activation of the nuclear transcription factor kappa B (NF-κB) [35,36]. It is well-known that NO production at the vascular level relies on two different mechanisms: an inducible synthase (iNOS) expressed upon the activation of a specific set of redox sensitive transcription factors, NF-kB possibly being the most important, and a constitutive one, the endothelial form eNOS, mainly regulated by Ca availability.

We have previously reported that variations in the composition and pressure of the air breathed are associated with a specific pattern of activation of oxygen-sensing transcription factors, including NF-kB, NRF2 and HIF [37]. In fact, the vasodilatory effect of superoxide dismutase in hyperoxia was not seen in animals given prior doses of the NO synthase inhibitor [34].

Despite the increase in the cardiac output, we observed a decrease in the mean arterial pressure during hypoxia. This can be explained by the decrease in peripheric vascular resistance. Unlike all other parameters, blood pressure does not come back to normal values one hour after returning to normoxic conditions [38]. This is probably a matter of time. Blood pressure should therefore be monitored for a longer period after an ascent to altitude.

Post-occlusive reactive hyperemia (PORH) remains unchanged throughout the experiment. Since the compensatory mechanisms aim to increase oxygen delivery, it is likely that hypoxia in the peripheral capillary bed is not sufficient to activate the mechanisms found during longer periods of hypoxia. A difference in reactivity between the macro and microcirculation is not new. Indeed, a study on divers showed that the response to exercise in the small arteries near the muscle is bigger than the variation in partial pressures of oxygen, leading to opposite effects on large arteries and peripheral circulation [3]. Later, another study demonstrated that the macro- and microcirculation could be affected differently [39]. Indeed, the shear stress experienced by the endothelium during the occlusion duration is lesser in small vascular conduits than in conductance arteries [40]. In addition, microcirculatory vessels could be more sensitive to the effect of pressure alone [41]. All of these mechanisms could explain why PORH does not change throughout the experiment while FMD varies.

Even though the purpose of the study was to observe the effects of a rapid ascent to altitude, the clinical implications of this study are multiple. First, the effects of permissive hypoxemia should be evaluated independently of clinical symptoms in clinical practice. An increase in oxygen delivery is the primary compensatory mechanism of permissive hypoxemia. The application of permissive hypoxemia should be evaluated in the context of patients’ capacity to increase cardiac output sufficiently to compensate for the decrease in hemoglobin oxygen saturation. Of note, close monitoring of heart function is mandatory as the increase in heart work and decreased NO production may cause cardiac dysfunction [42]. Second, hypoxemia-induced hemodynamic and vascular alterations are rapidly reversed after normoxia instauration. Third, an FMD decrease can be used as an index to further assess clinically significant hypoxemia. Future studies should evaluate whether FMD changes can be used as a tool for oxygen treatment titration in clinical practice for avoiding significant hypoxemia.

The strength of this study is that we investigated the hemodynamic effects of hypoxemia in healthy young human beings. The volunteers did not carry out physical activity during the study that could have further induced any local hypoxia or increased nitric oxide production—permitting us to investigate the physiological mechanisms of hypoxemia compensation. Furthermore, the degree of the hypoxemia and its installation velocity were progressive, which is a clinically relevant scenario, particularly in critically ill patients. However, this study has several limitations. First, hypoxemia was caused with the hypobaric method. Even though ambient pressure decreased slightly, we cannot exclude that it did not affect our results. Secondly, we assessed cardiac output with cardiac echocardiography, which is not a direct (invasive) method and therefore has its own limitations and uncertainties. Nevertheless, although the difference in CO between echocardiography by different types or sites and thermodilution was not entirely consistent, the overall effect of a meta-analysis showed that no significant differences were observed between ultrasound and thermodilution. In addition, the same investigator made all examinations, and analysis of the images was made blindly offline.

## 5. Conclusions

Acute exposure to hypobaric hypoxia resulted in decreased oxygen saturation and increased compensatory heart rate and cardiac output. The reduction in systemic vascular resistance is probably the explanation of the reduced mean blood pressure and the increased pre-occlusion vascular diameters preventing flow-mediated dilation to significantly increase at 3842 m. Post-occlusive reactive hyperemia did not change throughout the whole experiment, which shows that compensation mechanisms that increase oxygen delivery are effective. Almost all parameters return to normal values when coming back to normobaric conditions. Finally, even if the study observed the effects of an acute ascent in altitude, the clinical implications of this study are multiple.

## Figures and Tables

**Figure 1 ijerph-19-05394-f001:**
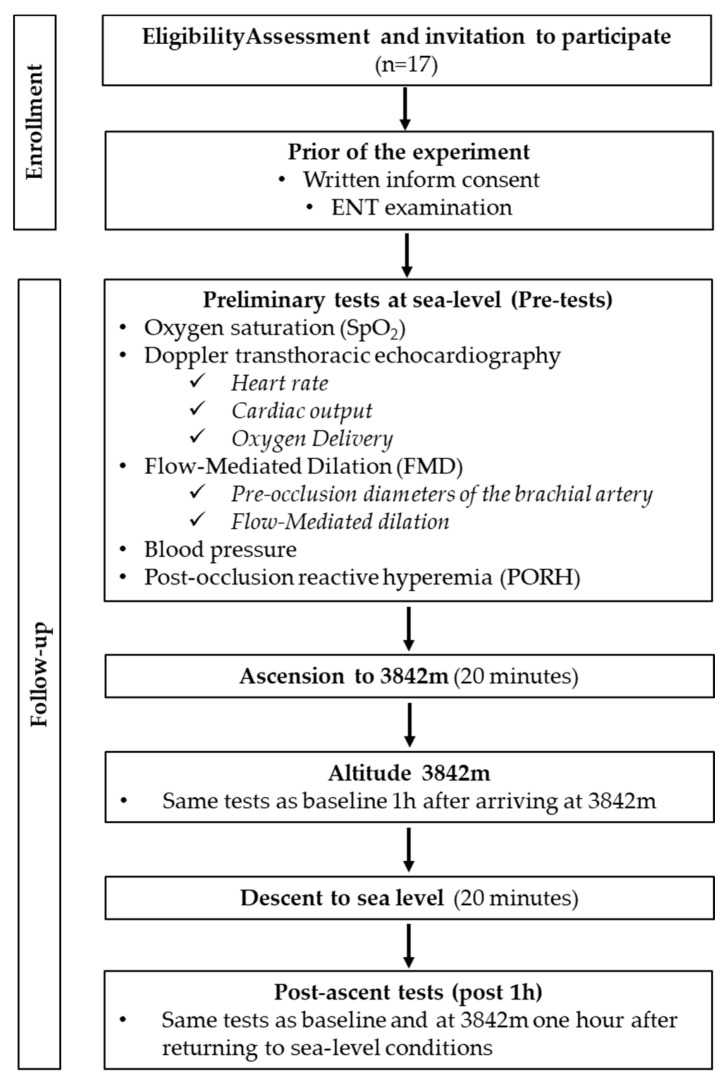
Flowchart of the experimental protocol.

**Figure 2 ijerph-19-05394-f002:**
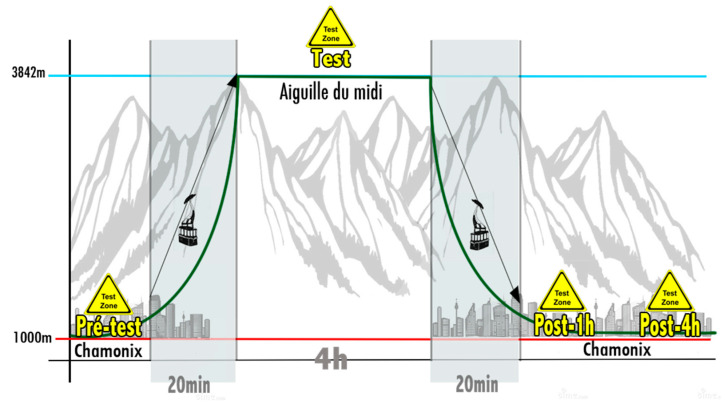
Changes in altitude during experiment. Measurements were taken at sea level, at 3842 m and 1 h after return to sea level.

**Figure 3 ijerph-19-05394-f003:**
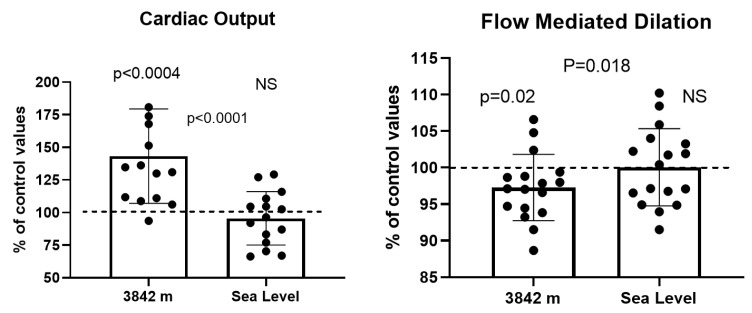
Cardiac output changes and flow-mediated dilation. Results are presented in relative values, at 3842 m and 1 h after return to normobaric conditions compared to pre-ascent values. Results are presented as mean ± SD. NS *p* > 0.05.

**Table 1 ijerph-19-05394-t001:** Absolute values of the saturation, cardiac and vascular parameters before ascent, at 3842 m and 1 h after return to normobaric conditions. Results are given in mean ± SD. *n* = 17 for all parameters except oxygen delivery index (*n* = 15), systemic vascular resistances (*n* = 12) and PORH (*n* = 14). Results are compared to pre-ascent values. * *p* < 0.05; ** *p* < 0.01; *** *p* < 0.001; ^ns^
*p* > 0.05.

	Pre-Ascent	3842 m	1 h Post-Descent	*n*
SpO_2_ (%)	97.7 ± 0.9	83.1 ± 4.2 ***	97.8 ± 0.9 ^ns^	17
**Cardiac parameters**				
Heart rate (bpm)	66 ± 15	81 ± 15 ***	65 ± 12 ^ns^	17
Oxygen delivery index	301.2 ±104.4	329.6 ± 81.6 *	266.3 ± 62.1 ^ns^	15
**Vascular parameters**				
Pre occlusion diameter (cm)	0.44 ± 0.04	0.46 ± 0.05 ***	0.45 ± 0.04 ^ns^	17
Systolic blood pressure (mmHg)	127.2 ± 8.3	118.7 ± 12.8 **	122.8 ± 18.0 *	17
Diastolic blood pressure (mmHg)	77.1 ± 7.6	71.4 ± 5.2 *	74.1 ± 9.5 ^ns^	17
Mean blood pressure (mmHg)	93.9 ± 6.8	87.2 ± 5.4 **	90.2 ± 11.6 *	17
Systemic vascular resistance (dynes.sec.cm^−5^)	2428 ± 1049	1786 ± 418.4 **	2726 ± 560 ^ns^	12
PORH (%)	133.4 ± 67.4	145.8 ± 77.2 ^ns^	116.1 ± 53.1 ^ns^	14

## Data Availability

Data are available at request from the authors.

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
