# Peer review of "Effects of Acute Hypobaric Hypoxia Exposure on Cardiovascular Function in Unacclimatized Healthy Subjects: A “Rapid Ascent” Hypobaric Chamber Study"

_ijerph, 2022, doi:10.3390/ijerph19095394_

Round 1
Reviewer 1 Report
The author aimed to exhibit the effects of a fast-acute ascent to simulated high altitude on cardiovascular function. Although this study is interesting, it has problems with how to present the study method, data analysis and results. Also, all the results of this study can be presented in the form of a table (Table 1) and the manuscript can be reported as a research letter.
Notes in the manuscript:
Endothelial function assays are written in the abstract and text of the article. Since the function of the endothelium has not been studied, it is better to use the term cardiovascular function.
Exclusion criteria should include any respiratory, cardiovascular, neurological and chronic diseases.
There is ambiguity about when the tests were taken after reaching the height. In Figure 1 it is written that the tests were performed 30 minutes after reaching the height, but in the first line of page 4 it is written 1 hour later and in Figure 2 it is written that the test was performed while they were at the height!!!
Why are the tests taken 30 minutes (or an hour) after reaching the height and one hour after returning?
In the analysis section of the manuscript, there appear to be written errors or the explanation of the statistical analysis method may not have been fully written. Comparison at altitude and after returning to sea level compared to before altitude (100%) should be done with one sample t-test or non-parametric equivalent. Comparison between parameters at altitude and after return to sea level should be done by paired t-test or non-parametric equivalent. Comparison between groups in Table 1 should also be done by RM one way ANOVA or non-parametric equivalent.
What additional information do the figures have compared to Table 1? For this manuscript, remove the figures and report only Table 1.
Author Response
Reviewer 1
The author aimed to exhibit the effects of a fast-acute ascent to simulated high altitude on cardiovascular function. Although this study is interesting, it has problems with how to present the study method, data analysis and results. Also, all the results of this study can be presented in the form of a table (Table 1) and the manuscript can be reported as a research letter.
Notes in the manuscript:
Endothelial function assays are written in the abstract and text of the article. Since the function of the endothelium has not been studied, it is better to use the term cardiovascular function.
The reviewer’ s point is taken, we only left “cardiovascular function” in the abstract and the text.
Exclusion criteria should include any respiratory, cardiovascular, neurological and chronic diseases.
Thank you for pointing this out, we added : “any respiratory, cardiovascular, neurological or chronic diseases” in the exclusion criteria. Nevertheless, all the subjects who partake in the study were considered healthy after an ENT medical examination as well as a questionnaire including, hay fever, lung problems, medications, heart problems, etc... were analysed by a medical doctor before starting the experiments.
There is ambiguity about when the tests were taken after reaching the height. In Figure 1 it is written that the tests were performed 30 minutes after reaching the height, but in the first line of page 4 it is written 1 hour later and in Figure 2 it is written that the test was performed while they were at the height!!!
Thank you for noticing it! Measurements were taken 1h after arriving at altitude. We uploaded a wrong version of the flow-chart. The figure has been changed accordingly.
Why are the tests taken 30 minutes (or an hour) after reaching the height and one hour after returning?
The measurements were made at altitude (see previous answer) to evaluate the effect of acute hypoxia on cardiovascular functions and one hour after the return to normoxia to observe how long these changes could perdure (or not) after returning to normal conditions.
In the analysis section of the manuscript, there appear to be written errors or the explanation of the statistical analysis method may not have been fully written. Comparison at altitude and after returning to sea level compared to before altitude (100%) should be done with one sample t-test or non-parametric equivalent. Comparison between parameters at altitude and after return to sea level should be done by paired t-test or non-parametric equivalent. Comparison between groups in Table 1 should also be done by RM one way ANOVA or non-parametric equivalent.
Thank you for your comment, what you pointed in your correction are indeed the tests that have been carried out. The Wilcoxon Rank sum test is the non-parametric equivalent of the one sample t test and comparison between parameters at altitude and after return to sea level has been omitted in this paragraph. We added it. The results in table 1 were indeed analyzed using the ANOVA test.
Again we thank the reviewer for helping us to improve our manuscript and we changed accordingly to add clarity: “The difference between the percentage of pre-ascension values and 100 % was compared by a two-tailed one-sample t test when normality of the sample was reached as assessed by the d’Agostino & Pearson test. Otherwise, the non-parametric Wilcoxon Rank Sum test was used. Comparison between parameters at altitude and after return to sea level were achieved by means of a paired t-test or Wilcoxon matched-pairs signed rank test when appropriate. Comparison between absolute values were done by Repeated Measures one-way ANOVA or Friedman test when normality was not assumed”.
What additional information do the figures have compared to Table 1? For this manuscript, remove the figures and report only Table 1.
The graphs show the relative evolution of the results in a graphical way whereas table 1 gives the absolute values of the results. In order to reduce the number of figures and avoid duplicate results, we left the graphs only for the 2 main parameters (cardiac output and FMD) and removed them from table 1.
Reviewer 2 Report
I feel very elated reading the manuscript by Theunissen et al.,. The manuscript is well-written and fall within the journal scope. The authors assessed short exposure to hypobaric hypoxia, comparable to the ascent to
the “Aiguille du Midi”, and how they could influence cardiovascular and endothelial function for unacclimatized tourists. This study is of public health importance and it add to general knowledge.
However I have some minor comments for authors.
Materials and Methods
The use of prospective observational study design is very broad. What type of observational study study was used. Authors must specify.
Eligibility criteria
Authors indicated that those who were age less than 50 years. Did they include a day year old baby? There must be a cut-off age group since the use of less than 50 is not specify knowing the mean age of 26 years.
Author Response
Reviewer 2
I feel very elated reading the manuscript by Theunissen et al.,. The manuscript is well-written and fall within the journal scope. The authors assessed short exposure to hypobaric hypoxia, comparable to the ascent to
the “Aiguille du Midi”, and how they could influence cardiovascular and endothelial function for unacclimatized tourists. This study is of public health importance and it add to general knowledge.
Thank you very much for your kind remark!
However I have some minor comments for authors.
Materials and Methods
The use of prospective observational study design is very broad. What type of observational study study was used. Authors must specify.
The design was a cross-sectional prospective study. We changed in the main text.
Eligibility criteria
Authors indicated that those who were age less than 50 years. Did they include a day year old baby? There must be a cut-off age group since the use of less than 50 is not specify knowing the mean age of 26 years.
Indeed ;-) We could have specified that the minimum age to participate in the study was 18 years old. However, the youngest subject was 20 years old. The sentence in the manuscript is misleading and we changed it as follows “….. were aged between 18 and 50 years old”.
Reviewer 3 Report
GENERAL COMMENTS
Thank you for the opportunity to review your paper. The topic of the paper is interesting and fits the scope of the journal. The text itself is well written and composed. However, there are some methodological issues that need to be clarified.
SPECIFIC COMMENTS
Section 2.1: Please consider reporting appropriate power analysis. Was a sample size of 17 sufficient for detecting statistical significance?
Section 2.2: Please consider reporting the four parameters describing the thermal components of the environment (temperature, humidity, wind, radiation). These environmental parameters can modify the cardiovascular strain experienced by your participants and thus it is important to be reported.
Section 2.3.2: “… this led to detection of 2 participants with clinically relevant dehydration; as the calculation could not be accepted, these two were therefore not considered for these measurements.” This indicates that the study involved monitoring 15 participants? If yes, please consider taking into account this sample size for the aforementioned power calculation.
Table 1 (legend): Please consider changing capital "N" (population) to small "n" (sample size) throughout the manuscript.
Discussion: “Unlike all other parameters, blood pressure does not come back to normal values one hour after returning to normoxic conditions.” Please consider strengthening this statement by citing published literature.
Discussion: “Secondly, we assessed cardiac output with cardiac echo-cardiography, what can be controversial.” Please consider rephrasing to improve clarity.
Author Response
Reviewer 3
GENERAL COMMENTS
Thank you for the opportunity to review your paper. The topic of the paper is interesting and fits the scope of the journal. The text itself is well written and composed. However, there are some methodological issues that need to be clarified.
SPECIFIC COMMENTS
Section 2.1: Please consider reporting appropriate power analysis. Was a sample size of 17 sufficient for detecting statistical significance?
With the kind of design provided (every subject being its own control, with relative measurements reported) and according to previous studies, power calculation is 95%. The following sentence was already in the manuscript: “Sample size was calculated setting the power of the study at 95%, and assuming that variables associated to altitude would have been affected similarly to what observed in previous studies [15]”.
Section 2.2: Please consider reporting the four parameters describing the thermal components of the environment (temperature, humidity, wind, radiation). These environmental parameters can modify the cardiovascular strain experienced by your participants and thus it is important to be reported.
This sentence has been added in the main text: “Temperature in the hypobaric chamber was 23°C and humidity 32%. There was no wind and no radiation”.
Section 2.3.2: “… this led to detection of 2 participants with clinically relevant dehydration; as the calculation could not be accepted, these two were therefore not considered for these measurements.” This indicates that the study involved monitoring 15 participants? If yes, please consider taking into account this sample size for the aforementioned power calculation.
See previous answer.
Table 1 (legend): Please consider changing capital "N" (population) to small "n" (sample size) throughout the manuscript.
Changed accordingly
Discussion: “Unlike all other parameters, blood pressure does not come back to normal values one hour after returning to normoxic conditions.” Please consider strengthening this statement by citing published literature.
We added this reference after the sentence:
Fox WC, Watson R, Lockette W. Acute hypoxemia increases cardiovascular baroreceptor sensitivity in humans. Am J Hypertens. 2006 Sep;19(9):958-63. doi: 10.1016/j.amjhyper.2006.02.005. PMID: 16942940.
Discussion: “Secondly, we assessed cardiac output with cardiac echo-cardiography, what can be controversial.” Please consider rephrasing to improve clarity.
Thank-you! In fact, we just wanted to point out that the measurement was not a direct one (invasive), and as such having its own limitations and uncertainties. We changed the sentence as follows: “. Secondly, we assessed cardiac output with cardiac echocardiography, which is not a direct (invasive) method and therefore has its own limitations and uncertainties”
Round 2
Reviewer 3 Report
Thank you for considering my suggestions and congratulations on presenting a wonderful study.
Author Response
Thank you for considering my suggestions and congratulations on presenting a wonderful study.
Thank you for suggesting improvements to this manuscript.